# CDKL5 Deficiency Disorder—A Complex Epileptic Encephalopathy

**DOI:** 10.3390/brainsci10020107

**Published:** 2020-02-17

**Authors:** Martyna Jakimiec, Justyna Paprocka, Robert Śmigiel

**Affiliations:** 1Department of Paediatrics and Developmental Age Neurology, Upper Silesian Child Health Centre, 40-752 Katowice, Poland; martyna.jakimiec@gmail.com; 2Department of Paediatric Neurology, Faculty of Medical Science, Medical University of Silesia, 40-055 Katowice, Poland; 3Department of Paediatrics and Rare Disorders, Wroclaw Medical University, 50-367 Wroclaw, Poland; robert.smigiel@umed.wroc.pl

**Keywords:** CDKL5 deficiency disorder, children, epilepsy

## Abstract

CDKL5 deficiency disorder (CDD) is a complex of clinical symptoms resulting from the presence of non-functional CDKL5 protein, i.e., serine-threonine kinase (previously referred to as STK9), or its complete absence. The clinical picture is characterized by epileptic seizures (that start within the first three months of life and most often do not respond to pharmacological treatment), epileptic encephalopathy secondary to seizures, and retardation of psychomotor development, which are often observed already in the first months of life. Due to the fact that *CDKL5* is located on the X chromosome, the prevalence of CDD among women is four times higher than in men. However, the course is usually more severe among male patients. Recently, many clinical centers have analyzed this condition and provided knowledge on the function of CDKL5 protein, the natural history of the disease, therapeutic options, and their effectiveness and prognosis. The International CDKL5 Disorder Database was established in 2012, which focuses its activity on expanding knowledge related to this condition and disseminating such knowledge to the families of patients.

## 1. Introduction

Mutations in the *CDKL5* gene should be considered in patients with epileptic seizures, and psychomotor and intellectual retardation. Epilepsy usually begins within the first three months of life and is often manifested by various seizures that respond poorly to antiepileptic treatment [1,2,3,4,5,6]. Vegetative disorders, hand stereotypy, poor eye contact, breathing disorders, sleep disorders, and gastrointestinal problems are less characteristic symptoms that should also be considered in diagnosis [7,8,9]. However, significant phenotypic variation in patients should be borne in mind. It may be related to the type and location of the mutation and the impact of epigenetic and environmental factors. Due to the limited therapeutic possibilities, patients with CDKL5 deficiency disorder (CDD) may experience permanent symptoms of epileptic encephalopathy and significant developmental impairment [4,8]. Better understanding of the natural history and the course of the disease will result in the development of molecular and genetic therapeutic possibilities. For patients and their families, such knowledge of the above disorders is of crucial significance in terms of genetic counseling.

## 2. History, Protein Function and Mutations

In 1985, Folker Hanefeld was the first to describe the early-onset seizure variant of Rett syndrome that preceded the diagnosis of intellectual disability [10]. In 2003, Kalscheuer [1] identified the locus of the gene, encoding a serine-threonine kinase (STK9) on the X chromosome (Xp22.3), and suggested the involvement of this gene mutation in the pathogenesis of X-linked infantile spasms (ISSX) based on the analysis of two cases of female patients with infantile spasms and profound mental retardation. In turn, Weaving in 2004 [2] and Scala in 2005 [11] confirmed the occurrence of mutations in the *CDKL5* gene in patients previously diagnosed with the Hanefeld variant of Rett syndrome and suggested its atypical form. Subsequent studies conducted on a larger scale showed significant and specific differences between CDD and syndromes whose atypical form was previously known as Rett syndrome [3,4,12]. Currently, CDD is treated as a separate disease entity. However, the clinical picture of the disease is often heterogeneous, which constantly motivates researchers to extend knowledge of the genetic background.

The *CDKL5* gene is located on the short arm of the X chromosome (Xp22). A protein belonging to the serine-threonine kinase family is the translation product [13]. It is a substance widely distributed in the human body with the highest expression in the brain (cerebral cortex, hippocampus, cerebellum, thalamus, and brainstem), the testicles, or the thymus [14]. The level of CDKL5 is different at various stages of development. It is lowest in the prenatal period and the highest concentrations are found in the peri- and postnatal stages of the rapid development of the nervous system (particularly in the cerebral cortex and the hippocampus) [5,15]. Changes in the level of CDKL5 protein suggest an important role in the process of neuronal formation and maturation. Although the exact molecular role of the protein has not been precisely defined, it is currently known that it is involved in proliferation, neuronal migration, neuronal formation, and neuronal growth, as well as in the development and functioning of synapses in brain maturation [5,7,16]. Studies show that it is involved in the phosphorylation of the product of MeCP2, whose mutations account for Rett Syndrome. This suggests a common metabolic pathway for both proteins, and therefore partly explains the clinical similarities of the conditions [2,5,11,15]. Intracellular changes in kinase concentration in cell compartments are also of critical importance to the proper function of neurons. Cytoplasmic CDKL5 determines the proper development of dendrites and dendrite bifurcation. However, the nuclear protein fraction is involved in RNA storage and modification, and splicing regulation [15]. Therefore, it determines the further normal structure, function, and survival of neurons. The CDKL5 protein is divided between regions, i.e., the ATP-binding region; the catalytic domain which determines the normal kinase function; and the long C-terminal domain, responsible for the regulation of catalytic activity of the protein and intracellular distribution of the kinase through active nuclear transport [6,17,18]. Studies show that patients with mutations within the catalytic domain and frameshift mutations located at the end of the C-terminal region had more severe motor impairment, refractory epilepsy, or microcephaly [17]. Milder forms were observed in patients in whom mutations within the ATP-binding region or nonsense mutations in the C-terminal region were identified. In such patients, better hand use and the ability to walk unaided were reported [7,18].

So far, more than 265 pathogenic variants within the *CDKL5* gene have been reported. About 50% of these variants are point mutations. Missense mutations are the most common (38%). However, only 27% of them are considered pathogenic and many mutations are related to the catalytic domain. It was demonstrated that the frameshift mutation (due to point insertion or deletion) is the most pathogenic mutation in the *CDKL5* gene. Such variants are present in approximately 13% of patients [5,19]. In addition to point mutations, studies reported many cases of deletion of *CDKL5* (several or all exons), and even a significant part of the short arm of the X chromosome, containing many genes. The clinical picture of the above cases was not significantly different from those caused by point mutations [17,20,21,22,23].

Although most cases of CDD are the result of de novo mutations, cases of family history of *CDKL5* mutations were also reported. This is probably the effect of germline mosaicism in one of the parents. Therefore, in these situations the offspring can inherit the mutated gene. Noteworthy also is phenotypic divergence among siblings in whom the same mutation was confirmed. This suggests the involvement of epigenetic and environmental factors, as well as the relationship with the inactivation of the X chromosome, in determining the final phenotypic picture of particular patients [2,24,25,26]. The frequency of mosaicism variants is estimated at about 8.8% of CDDs [27]. Therefore, it seems reasonable to provide genetic counseling to couples whose child is affected with CDD or offer an extension of prenatal diagnosis, including a mutation of the *CDKL5* gene in subsequent pregnancies.

## 3. Clinical Picture of the Disease

The prevalence of mutations in *CDKL5* is estimated at approximately 1 in 40,000–60,000 live births. The primary symptoms of CDD include early-onset epilepsy (mostly drug-refractory), generalized hypotonia, psychomotor developmental disorders, intellectual disability, and cortical vision disorders. In addition, a number of accompanying symptoms are reported, e.g., autistic features—poor social interactions, poor eye contact; hand stereotypy, vegetative disorders, gastrointestinal and orthopedic complaints, or dysmorphic facial features [5,7,23]. The clinical picture of CDKL5 disorders is not uniform.

CDD is linked to the X chromosome and affects the female gender four times more often than men, which suggests that it is mostly a lethal mutation in male fetal life. Inactivation of the X chromosome seems to be one of the most significant factors determining the severity of symptoms in the female gender [12,28]. The *CDKL5* mutation is diagnosed in 8–16% of girls with early-onset epilepsy [3,5]. However, in the population with early infantile epileptic encephalopathy (EEIE), this mutation was confirmed in 28% of girls and only in 5.4% of boys [29]. The number of affected boys confirms the fact that the course of the disease is much more severe in boys, and often results in death in the first or second decade of life [25].

Phenotypically, the disease is diverse not only in relation to gender. Studies report cases ranging from mild to severe forms of epileptic encephalopathy. Mild forms, however, are significantly less prevalent. In those cases, patients walk unaided, use simple sentences, and epilepsy is controlled pharmacologically. Severe forms of epileptic encephalopathy include attacks not responsive to drug therapy, microcephaly, profound mental and psychomotor retardation, generalized hypotonia, and cortical vision disorders. Despite extensive studies in this respect, the correlation between the type or location of the mutation and the severity of symptoms cannot be clearly determined.

Epileptic seizures are usually the first symptom of CDD. In 96.9% of patients, the first episodes are found in the first six months of life, and in the first three months of life in 90% (usually week 6) [7,30]. Early manifestation and diagnosis have a huge impact on the quality of life of patients and their families, as well as on psychomotor and intellectual development. Epileptic episodes may have different morphology. Epileptic spasms usually occur at the onset of the disease in 23% of patients and at various stages of the disease in 81% of patients. Other possible types include absence, partial, myoclonic, tonic seizures, and limb spasms [31,32]. Over time, seizures become generalized or multifocal with the most prevalent tonic-clonic manifestations. Recently, a new model of epileptic seizure evolution has been proposed as a result of the analyses of the morphology of seizures in children with CDD. Attention was paid to three phases: hypertonic (intense movements, muscle jerks), tonic, and epileptic. This seizure model was reported in 56% of subjects [5,7,33,34,35,36,37].

Epileptic seizure does not occur in only about 1.74% of patients. These cases are very rare. In other patients who experience epilepsy < 10% of patients have several episodes a month, about 12% have several episodes weekly and up to 80% have seizures every day. In the last group of patients, up to five seizures per day were noted in 68% of subjects, 5–10 daily in 22%, and more than 10 in 10% of patients. Patients with 20 epileptic attacks per day were reported. The average number of attacks was two daily among all patients [8,30]. Epilepsy in CDD shows high drug resistance (84%) [38]. Frequent and intense epileptic episodes significantly affect the psychomotor and intellectual development of children, and result in epileptic encephalopathy, which has a secondary effect on the functioning and the quality of life of patients and their families [39].

Retardation of psychomotor development and intellectual disability affect all patients with CDD. In almost all patients, intellectual disability is indicated at a moderate or more often significant degree. Therefore, patients achieve an IQ score of no more than 40 [38]. The impairment seems to be mainly related to speech and small motor skills compared to achievements in gross motor skills [5]. The analysis of the achievement of milestones by CDD patients shows that boys are mostly affected.

Nearly 66% of girls with CDD can sit unsupported, 25% can stand, 21% can stand up from the sitting position, 23% of patients walk unaided until the age of 4.5 years, and 13% of patients can run at any stage of development. However, only about 35% of examined boys can sit unsupported. Isolated cases of affected boys who were able to stand (2/18; 11%) or walk independently (1/18; 5.5%) were reported [5,8,9,20,38,40]. About 50% of the girls achieve grasping ability up to five years of age and 13% of girls achieve pincer grip at any stage. Boys show poorer development of small motor skills, only 10% develop grasping ability up to two years of age, and only one boy achieved pincer grip at that time [9].

Speech disorders in patients with the *CDKL5* mutation are more severe than those related to gross motor skills. Studies indicate that 44% of patients babbled up to the age of six, and only 16% of patients were able to use single words up to seven years of age. In turn, only 7.5% of girls used simple sentences (in the literature, one case of a boy able to speak single words was also reported). Symbols, gestures, facial expressions, or vocalizations are the most common methods of communication used by about 65% of girls. In boys, only one in four communicated using gestures or vocalizations [2,3,5,8,9,20,38].

Despite severe psychomotor retardation and significant intellectual disability, developmental regression is rarely observed in CDD patients. It affects approximately 20% of patients. This is the feature differentiating CDD from Rett syndrome, in which the loss of previously achieved milestones is a typical symptom. Eighty percent of patients with CDD present with retardation or inhibition of psychomotor development. Regression is reported in the cases of severe epileptic encephalopathy and is usually related to communication skills and manual dexterity [9].

Generalized hypotonia affects all patients. However, it is more severe in boys [8,41]. Scoliosis is its relatively common orthopedic complication in children with CDD and occurs in 8.3% of patients. The probability of developing scoliosis before the age of 10 is 68.5%. Rehabilitation, orthopedic corsets and surgical treatment are used to treat scoliosis [7,30,38].

Cortical visual disturbances are common and occur in 80% of patients. They manifest clinically as lack of fixation and lack of following objects with the eyes. Avoidance of eye contact is observed in 40.6% of patients, which is also interpreted as one of the autistic features. Less commonly reported symptoms include rotational or horizontal nystagmus [3,5,17,23,33,38].

Hand stereotypies occur in 85.7% of patients and are one of the most common accompanying symptoms. Their prevalence is six-fold higher in girls and reaches 91% compared to 15% in boys. Putting hands in the mouth and clapping are most commonly observed. Symptoms usually appear in the first year of life and are more intense over time [5,38,42]. Crossing lower limbs is also common. Cases of episodic choreoathetosis, akathisia, and dystonia were also reported [7].

The head circumference at birth is normal in most patients (93.8%), but the inhibition of head circumference growth is observed in the first years of life. In 44.4% of patients with CDD this size indicates the result of <3 percentile as early as at the age of two [38].

Vegetative disorders are a diverse group of symptoms associated with CDD. These include respiratory disorders (apnea or hyperventilation), gastrointestinal disorders, dilated pupils, facial flushing, or cooling of the limbs. These symptoms are almost three times more prevalent among the female gender [5,7,38].

Studies show that 86.5% patients experience at least one episode of gastrointestinal disorder [30]. 71% of patients are affected by constipation, 63% have symptoms of gastroesophageal reflux disease, and 27% experience aerophagia. Gastrointestinal disorders are almost four times more prevalent in patients over five years of age compared to younger children [30].

Significant problems are observed in relation to nutrition. They affect almost all patients [38]. 79.3% are orally fed, of whom only 5.3% of patients are independent in this respect. At least one episode of choking per week during feeding is found in almost 75% of orally fed patients, which drastically increases the risk of aspiration and pneumonia. Due to feeding and swallowing disorders, 20.7% of patients are given enteral nutrition (gastrostomy or gastric tube). Gastrostomy is more common in boys, as they experience more severe difficulties in nutrition and require earlier introduction of enteral feeding methods (25% of boys aged 2.5 years) [30]. Of note, over 40% of patients show body weight below the 25th percentile [38].

Respiratory disorders occur in one in three patients with CDD and mostly include hyperventilation (13.6%) and apnea attacks (26.3%). In addition, 21.4% of patients have a history of pneumonia. Aspirations, which affect 22.6% of patients, are also a serious problem. It was demonstrated that the male gender is more at risk of respiratory disorders [30].

Sleep disorders are common in approximately 86.5% of patients with CDD, of whom 58.5% present with somnambulism. However, the entire spectrum of sleep disorders is observed (problems with falling asleep, awakening, breathing disorders during sleep, difficulty waking up, or pathological drowsiness). The highest intensity of symptoms is observed in patients aged 5–10 years [30].

Discreet dysmorphic features are observed in a small number of patients (5.7%) and they mostly include deep set eyes, straight eyebrows, slightly short upturned nose, large ears, high forehead, or prominent lips [7,12].

Other less commonly reported symptoms include bruxism (44%) and arrhythmias, hypersensitivity to touch, apraxia, or tetraplegia. It should be of note that CDD and many other syndromes characterized by frequent epileptic seizures may carry the risk of Sudden Unexpected Death in Epilepsy (SUDEP) [7,38,39].

## 4. Imaging Studies

Brain MRI images in the first months of life rarely show any abnormalities. Approximately 50% of 6-year-old children present with changes in neuroimaging. However, these are not very specific and are mostly related to (frontal) brain atrophy and white matter enhancement, mainly involving temporal lobes. Some patients also presented with wide sulci, widening of the ventricular system, or cerebellar atrophy. These changes are of different severity and occur in many other conditions. Therefore, they cannot be used as the basis for diagnosis, differentiation, or prognosis. It should be noted that the frequency of abnormal neuroimaging (as well as the severity of clinical symptoms) is also gender dependent. In 52.9% of examined boys, symptoms of brain atrophy were found in MRI, while this phenomenon was almost twice as less frequent in girls (25.7%) [3,6,7,12,17,23,36].

## 5. Therapy

Currently, there is no targeted therapy which could solve the underlying problem related to CDKL5 disorders. Therapeutic methods in patients with CDD are based on symptomatic drug use to control the most problematic complaints that increase disability, and to increase the possibility for development of individual patients. In addition, patients should be provided with multidisciplinary care, including physiotherapy, occupational therapy, neurological speech therapy, and dietetics [5].

It seems that epilepsy remains the area in which the greatest therapeutic effects could be achieved. The choice of an antiepileptic drug should be based on the current seizure type and EEG recordings. However, it should be borne in mind that resistance to antiepileptic drugs in CDD reaches 84%. It was found that the frequency of epileptic seizures decreased by 50% in 69% of patients in the third month after the beginning of antiepileptic therapy; in 45% in the sixth month; and only in 24% after one year. The most significant effects were initially noted in patients on valproate, lamotrigine, vigabatrin, clobazam, zonisamide, felbamate, or steroids. However, the effectiveness of these agents decreased over time. In view of the above data, multiple drug therapy is used in most cases. Only about 11% of patients do not require antiepileptic treatment, single drug therapy is used in 16% of patients, 28% of patients are on two drugs, 29.5% on three drugs, 14% require four drugs and 1.5% of patients require five drugs [43]. Other studies found that in only 43.6% of subjects was complete seizure control achieved for more than two months. The seizure-free period after introduction of new treatment is known as the “honeymoon”. It was noted that it occurs most frequently at the age of two and lasts, on average, six months (from several weeks to six years) [5,7,8,40,43].

Studies have been conducted on the effectiveness of cannabidiol therapy in patients with mutations in the *CDKL5* gene and other genetic epileptic encephalopathies. In those patients the prevalence of seizures decreased by more than 41% after 12 weeks of therapy and by almost 53% after 48 weeks of treatment [44].

In addition, therapies based on nonpharmacologic methods are used. They include ketogenic diet, vagus nerve stimulation (VNS), or callosotomy [45]. Ketogenic diet was used in approximately 15% of patients. The reduction of epileptic seizures after the inclusion of the diet was initially related to 58.7% of patients. However, adverse effects (gastrointestinal complaints or kidney stone disease) were contraindications to the diet in the case of about 33% of patients. Problems were also related to difficulties in following strict diet recommendations. Despite good initial effects, the frequency of seizures increased over time in most patients. Of note, patients with *CDKL5* mutations constitute the group with lowest effectiveness of ketogenic diet treatment among other genetic epileptic encephalopathies [46,47]. Callosotomy and VNS are rarely used invasive approaches to epilepsy in CDD. A reduction in seizures was observed in 69% of patients undergoing VNS [48,49]. No studies have been published to confirm the effectiveness of callosotomy. However, isolated cases of epileptic seizure reduction have been described in the International CDKL5 Disorder Database [7]. In addition, preparations aimed at reducing the accompanying symptoms are used as adjuncts to therapy (e.g., melatonin in sleep disorders). Gastrostomy or gastric probe are the methods of nutrition of patients that should also be noted [30].

Studies have been conducted to develop precise therapy based on the biological, metabolic and genetic basis of CDD. These studies include the use of NMDA (*N*-methyl-D-aspartate) receptor modulators [50]; allopregnanolone (a neurosteroid that restores normal microtubule morphology) [51,52]; tianeptine, which is an antidepressant affecting AMPA (α-amino-3-hydroxy-5-methyl-4-isoxazolepropionic acid) receptors [53]; or insulin-like growth factor IGF-1 activating the Akt/mTOR pathway [54]. Hopes are related to gene therapy as a causal treatment for disorders due to *CDKL5* mutations.

Research is constantly being conducted to develop precise therapies based on biological, metabolic and genetic bases for CDD. These studies include the use of NMDA receptor modulators: allopregnanolone (a neurosteroid restoring normal morphology of microtubules); tineptin (an antidepressant affecting AMPA receptors); and insulin-like growth factor IGF-1, activating AKT/mTORY serotonergic receptor agonist 5-HT7R—LP-211 [55,56]. Recently, the results of a study carried out on a CDD-loaded mouse population have been published, which demonstrated the beneficial effect of GSK3β receptor (tideglusib) inhibition on hippocampal development and hippocampus-dependent learning memory in young individuals [57]. The achievements of modern technology also allow the development of methods of protein substitution therapy, which was described by Trazzi [58]; TAT-CDKL5 fusion proteins were used, which, crossing the barrier, reach the CNS while maintaining the CDKL5 protein activity. The greatest hopes are placed in gene therapies as a form of causal treatment of disorders caused by CDKL5 mutations; an important discovery in this respect is Balester’s study, which demonstrated that in some cases the method using U1snRNA-mediated splicing correction may fully restore CDKL5 protein synthesis, subcellular distribution and kinase activity [59]. Ataluren’s therapeutic effects on CDD are also currently being studied. The above publications cast new light on the therapeutic possibilities in CDD.

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
