# Peer review of "CDKL5 Deficiency Disorder—A Complex Epileptic Encephalopathy"

_brainsci, 2020, doi:10.3390/brainsci10020107_

Round 1

Reviewer 1 Report

This manuscript by Jakimiec et al. is an excellent review of the clinical aspects of the CDKL5 Deficiency Disorder (CDD).

The authors have extensively reviewed existing publications on CDD, and have extracted pivotal data accurately. This review would help readers grasp the whole picture of the wide spectrum of this disorder. I have only a few suggestions, otherwise it is a well-done piece of work.

Suggestions:

1. Line 86. Duplications of the CDKL5 gene cause different type of disorder than the CDD, and do not cause an epileptic encephalopathy. “duplications” should be deleted, or described as such, citing papers such as “Szafranski, P., et al. Neurodevelopmental and neurobehavioral characteristics in males and females with CDKL5 duplications. European journal of human genetics : EJHG 23, 915-921 (2015).”

2. The human CDKL5 gene should be italicized when published, whereas the CDKL5 protein is not italicized.

Author Response

Dear Reviewer,

Thank you very much for extensive reviews and valuable sugestions.

„Duplications of the CDKL5 coding region”is deleted from the line 86.

Referrences to the gene “CDKL5” are corrected in italic.

Reviewer 2 Report

The paper by Jakiemiec et al. “CDKL5 deficiency disorder – a complex epileptic encephalopathy” highlights the most recent clinical findings on CDKL5 deficiency disorder. Although the clinical findings are very well summarized, the paper lacks some information and comments that should be added prior to publication.

In the abstract the authors state “Due to the limited therapeutic possibilities, patients with CDD may experience permanent, increasing symptoms of epileptic encephalopathy.” Actually, to my knowledge, there is no scientific evidence that CDD-symptoms worsen with age. Please reformulate the phrase or add a comment. The overall paper lacks correct citations. For example in paragraph 2 “History, protein function and mutations” important citations are missing.

               - “The level of CDKL5 is different at various stages of development. It is the lowest in the prenatal period, and the highest concentrations are found in the peri- and postnatal stages  of the rapid development of the nervous system (particularly in the cerebral cortex and        the hippocampus).” (CIT. Rusconi et al. 2008).

               - “Changes in the level of CDKL5 protein suggest an important role in the process of neuronal formation and maturation. Although the exact molecular role of the protein has not been precisely defined, it is currently known that it is involved in proliferation, neuronal migration, neuronal formation, and neuronal growth, as well as in the development and functioning of synapses in the brain maturation.”  Please insert correct citations on the papers which refer to these findings.

The above mentioned parts are examples and the whole paper should be revised carefully, in order to add correct citations.

The conclusions on different studies on therapies are incomplete. Several therapeutic approaches, including GSK3-beta inhibitors, molecules targeting serotonergic dysfunction are not mentioned. In addition, a comment on protein replacement therapy should be added. A comment on the actually undergoing clinical trials for CDD would also be appropriate.

Please review carefully grammatical and stylish errors. For example, when referred to the gene, “CDKL5” should be written in italic.  

Author Response

Dear Reviewer,

Thank you very much for extensive review and valuable sugestions.

1. “Due to the limited therapeutic possibilities, patients with CDD May experience permanent, increasing symptoms of epileptic encephalopathy.”-the sentence is corrected

2. Correct citations are added in lines: 30, 32, 36, 54, 56, 59, 64, 66, 70, 77, 110, 143, 194, 269, 272.

3. New therapies and on-going researches including protein replacement therapy, CSK3-beta inhibitors, molekule stargeting serotonergic dysfunction, are complemented in the last paragraph.